# Towards Aligned Layout Generation via Diffusion Model with Aesthetic Constraints

**Jian Chen**[1], **Ruiyi Zhang**[2], **Yufan Zhou**[2], **Rajiv Jain**[2], **Zhiqiang Xu**[3]
**Ryan Rossi**[2], **Changyou Chen**[1]
University at Buffalo[1], Adobe Research[2], MBZUAI[3]
`{jchen378,changyou}@buffalo.edu, ruizhang@adobe.com`

## Abstract

Controllable layout generation refers to the process of creating a plausible visual arrangement of elements within a graphic design (*e.g.*, document and web designs) with constraints representing design intentions. Although recent diffusion-based models have achieved state-of-the-art FID scores, they tend to exhibit more pronounced misalignment compared to earlier transformer-based models. In this work, we propose the **LA**yout **C**onstraint diffusion mod**E**l (LACE)[1], a unified model to handle a broad range of layout generation tasks, such as arranging elements with specified attributes and refining or completing a coarse layout design. The model is based on continuous diffusion models. Compared with existing methods that use discrete diffusion models, continuous state-space design can enable the incorporation of differentiable aesthetic constraint functions in training. For conditional generation, we introduce conditions via masked input. Extensive experiment results show that LACE produces high-quality layouts and outperforms existing state-of-the-art baselines.

## 1 Introduction

Leveraging advanced algorithms and artificial intelligence, automated layout generation serves as a cost-effective and scalable tool that facilitates a diverse range of applications including website development (Pang et al., 2016), UI design (Deka et al., 2017), urban planning (Yang et al., 2013), and editing in printed media (Zhong et al., 2019). Layout generation tasks fall into either unconditional or conditional categories. Unconditional generation refers to the process where layouts are generated from scratch without predefined conditions or constraints. Conditional generation, on the other hand, is guided by user specification, such as element types, positions and sizes, or unfinished layouts, allowing for more controlled and targeted results.

Previous research utilized generative models such as GANs (Goodfellow et al., 2014; Kikuchi et al., 2021), VAEs (Kingma & Welling, 2013), and transformer-based (Vaswani et al., 2017; Kong et al., 2022) models. Recent studies, however, have shifted the focus towards the application of diffusion models for better generative quality and versatility in conditional generation (Hui et al., 2023).

Diffusion-based models handle layout attributes as discrete or continuous variables and corrupt the data using categorical and Gaussian noise based on the discrete (Austin et al., 2021) and continuous (Ho et al., 2020) diffusion frameworks. These different corruption mechanisms result in distinct patterns during the generation process, as illustrated in Figure 1. Discrete diffusion, starting from a blank canvas, generates elements incrementally. Emerged elements tend to remain static, thus limiting the model's ability to make global adjustments. Continuous diffusion, by contrast, starts with a random arrangement and refines it to an organized one over time, which is arguably more flexible in modeling.

Diffusion models, while achieving state-of-the-art FID scores, usually underperform in terms of alignment and MaxIoU scores compared to earlier transformer-based models (Gupta et al., 2021), especially in unconditional generation (Inoue et al., 2023). These metrics can be leveraged for constraint optimization in continuous diffusion models to enhance layout aesthetics. However, due to

---

[1] Code is available at https://github.com/puar-playground/LACE

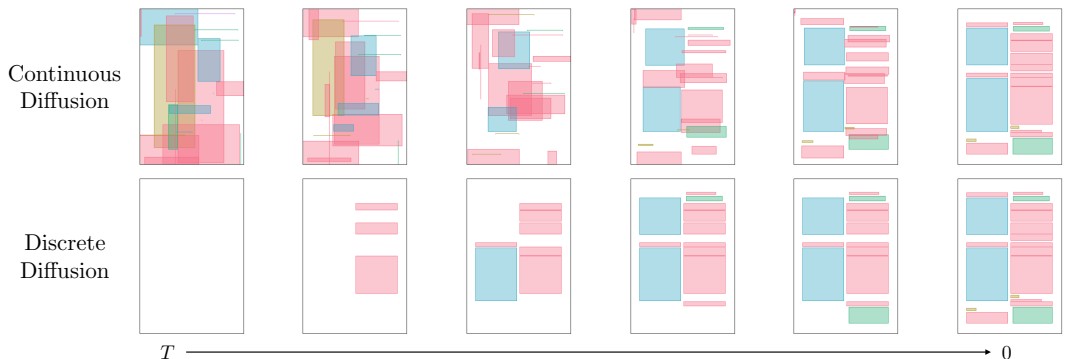

Figure 1: Comparisons of latent states in continuous and discrete diffusion for layout generation. Discrete diffusion adds elements to a blank canvas incrementally, and the added elements remain fixed. Continuous diffusion gradually refines a random layout into an organized one over time.

the non-differentiability of quantized geometric attributes, discrete models miss out on this optimization potential. On the other hand, continuous diffusion models face challenges in task unification because the sample spaces of Gaussian distribution and the data distribution (canvas range and probability simplex) are different. In contrast to discrete diffusion models (Inoue et al., 2023; Hui et al., 2023), which can direct unconditional models to meet conditional goals by masking specific attributes, this strategy proves inefficient for continuous models. This limitation could explain why earlier research on continuous diffusion models (Chai et al., 2023; He et al., 2023) focused on a single layout generation task.

In this study, we introduce the LAyout Constraint Diffusion modEl (LACE). A unified model designed to generate both geometric and categorical attributes for various tasks in a continuous space. While the neural network is trained to predict noise following the classic DDPM framework (He et al., 2023), we employ a simple reparameterization technique to compute a layout prediction (Song et al., 2020). This prediction then serves as a target to apply differentiable aesthetic constraint functions, thereby enhancing the model performance. In addition, we designed a global alignment loss and a pairwise overlap loss that serve as constraint functions during the training and post-processing stages. Global alignment loss promotes the learning of alignment patterns from real data during training and is used to refine global alignment in the post-processing stage. We train models with masked input to unify unconditional and conditional generation, following an approach similar to masked autoencoders, as seen in DiffMAE (Wei et al., 2023). To avoid convergence to the local minimum introduced by the constraints, we propose a time-dependent weight to deactivate the constraints for noisier time steps.

Our contribution is summarized as:

- Built upon a diffusion model, we formulate various controllable layout generation tasks as conditional generation processes in continuous space, enabling constraint optimization for enhanced quality.

- We propose two aesthetic constraint loss functions that promote global alignment and minimize overlap in the layout. These functions serve as constraints during both the training and post-processing phases.

- We conducted extensive experiments and achieved state-of-the-art results on public benchmarks across various layout generation tasks.

## 2 METHODOLOGY

### 2.1 PRELIMINARY: CONTINUOUS DIFFUSION MODELS

Diffusion models (Ho et al., 2020) are generative models characterized by a forward and reverse Markov process. In the forward process, a continuous valued data-point $\mathbf{x}_0$ is gradually corrupted to intermediate noisy latents $\mathbf{x}_{1:T}$, which converges to a random variable of the standard multi-variant Gaussian distribution $\mathcal{N}(0, \mathbf{I})$ after $T$ steps. The transition steps between adjacent intermediate predictions is modeled as Gaussian distributions, $q(\mathbf{x}_t|\mathbf{x}_{t-1}) = \mathcal{N}(\mathbf{x}_t; \sqrt{1 - \beta_t}\mathbf{x}_{t-1}, \beta_t\mathbf{I})$, with

a variance schedule $\beta_1, \cdots, \beta_T$. The forward process admits a closed-form sampling distribution, $q(\mathbf{x}_t|\mathbf{x}_0) = \mathcal{N}(\mathbf{x}_t; \sqrt{\bar{\alpha}_t}\mathbf{x}_0, (1-\bar{\alpha}_t)\mathbf{I})$ at arbitrary timestep $t$, where $\bar{\alpha}_t = \prod_{s=1}^{t} \alpha_s$ and $\alpha_t = (1-\beta_t)$.

In the reverse (generative) process, the data $\mathbf{x}_0$ is reconstructed gradually by sampling from a series of estimated transition distributions $p_\theta(\mathbf{x}_{t-1}|\mathbf{x}_t)$ start from a standard Gaussian random variable $X_T \sim \mathcal{N}(0, \mathbf{I})$. The transition distributions are learned by optimizing the evidence lower bound consists of a series of KL-divergence:

$$\mathbb{E}_q \left[ D_{\text{KL}}\left(q(\mathbf{x}_T|\mathbf{x}_0)||p(\mathbf{x}_T)\right) + \sum_{t>1}^{T} D_{\text{KL}}\left(q(\mathbf{x}_{t-1}|\mathbf{x}_t, \mathbf{x}_0)||p_\theta(\mathbf{x}_{t-1}|\mathbf{x}_t)\right) - \log p_\theta(\mathbf{x}_0|\mathbf{x}_1) \right]. \quad (1)$$

Since the posterior distributions $q(\mathbf{x}_{t-1}|\mathbf{x}_t, \mathbf{x}_0)$ are also Gaussian for an infinitesimal variance $\beta_t$ (Feller, 2015) and can be approximated by $\mathbf{x}_0$-parameterization (Song et al., 2020) as:

$$q(\mathbf{x}_{t-1}|\mathbf{x}_t, \mathbf{x}_0) = \mathcal{N}(\mathbf{x}_{t-1}; \tilde{\boldsymbol{\mu}}(\mathbf{x}_t, t), \tilde{\beta}_t\mathbf{I}) \quad (2)$$

$$\text{where} \quad \tilde{\boldsymbol{\mu}}(\mathbf{x}_t, t) = \frac{1}{\sqrt{\alpha_t}}\left(\mathbf{x}_t - \frac{\beta_t}{\sqrt{1-\bar{\alpha}_t}}\boldsymbol{\epsilon}\right), \ \boldsymbol{\epsilon} \sim \mathcal{N}(0, \mathbf{I}) \ \text{ and } \ \tilde{\beta}_t = \frac{1-\bar{\alpha}_{t-1}}{1-\bar{\alpha}_t}\beta_t, \quad (3)$$

the transition distributions could be estimated as a Gaussian with a mean:

$$\boldsymbol{\mu}_\theta(\mathbf{x}_t, t) = \frac{1}{\sqrt{\alpha_t}}\left(\mathbf{x}_t - \frac{\beta_t}{\sqrt{1-\bar{\alpha}_t}}\boldsymbol{\epsilon}_\theta(\mathbf{x}_t, t)\right), \quad (4)$$

where $\boldsymbol{\epsilon}_\theta$ is a function approximator and is usually trained using a simplified objective:

$$\mathcal{L}_{\text{simple}} = ||\boldsymbol{\epsilon} - \tilde{\boldsymbol{\epsilon}}_\theta(\sqrt{\bar{\alpha}_t}\mathbf{x}_0 + \sqrt{1-\bar{\alpha}_t}\boldsymbol{\epsilon}, t)||^2, \quad (5)$$

where $\boldsymbol{\epsilon} \sim \mathcal{N}(0, \mathbf{I})$.

## 2.2 CONTINUOUS LAYOUT GENERATION

Similar to previous work (Inoue et al., 2023; Chai et al., 2023; He et al., 2023), we define a layout with $l$ elements as $\mathbf{x} = \{(c_1, \mathbf{b}_1), ..., (c_l, \mathbf{b}_l)\}$, where $c_i \in \{0, 1, \cdots, N-1\}$ and $\mathbf{b}_i \in [0, 1]^4$ is the label (spanning $N$ classes) and the bounding box for the $i$-th element. The bounding box is defined by its center coordinates (x, y) and sizes ratio (width, height). In order to allow variable length generation, we extend layouts to a uniform length $L$ with padding elements, which has an extra class label $c = N$ and a bounding box initialized to zero, denoted as $b = (0, 0, 0, 0)$. Consequently, any layout within a dataset of $N$ classes is represented as a vector sequence with $N + 5$ dimensions and a length of $L$.

In contrast to previous discrete diffusion (Austin et al., 2021) based methods that quantize bounding box attributes to discrete bins, our approach considers them as continuous size and position ratio ranging from 0 to 1. This shift towards continuous variables enables the integration of continuous constraint functions to optimize aesthetic qualities. Moreover, following previous work (Han et al., 2022), we employ continuous label vectors of $(N + 1)$ dimension to represent noisy classification logits. In other words, label vectors are not confined in the probability simplex when $t > 0$. However, the clean data at $t = 0$ still has a one-hot label vectors. This design allows direct application of the classic diffusion model using Gaussian noise (Ho et al., 2020) without modality-wise corruption tricks (Hui et al., 2023; Inoue et al., 2023). An example is shown in Figure 2.

**Conditional Generation** Instead of training separate models for unconditional and various conditional generation, we train a single neural network to handle multiple generation tasks. In training, we employ three types of binary condition masks as data augmentation that fix the label, size attributes of all elements or all attributes of partial elements. Specifically, given a binary condition mask $\mathbf{m}$, the noisy latent is augmented as $\hat{\mathbf{x}}_t = \mathbf{m} \circ \mathbf{x}_0 + (\mathbf{1} - \mathbf{m}) \circ \mathbf{x}_t$, where $\mathbf{1}$ denotes a all-ones matrix.

## 2.3 RECONSTRUCTION AND AESTHETIC CONSTRAINTS

Following recent work, we introduce a reconstruction loss to encourage plausible predictions of $\mathbf{x}_0$ at each time step (Austin et al., 2021; Hui et al., 2023). Thus, the total loss is defined as $\mathcal{L} = \mathcal{L}_{\text{simple}} + \mathcal{L}_{\text{rec}}$.

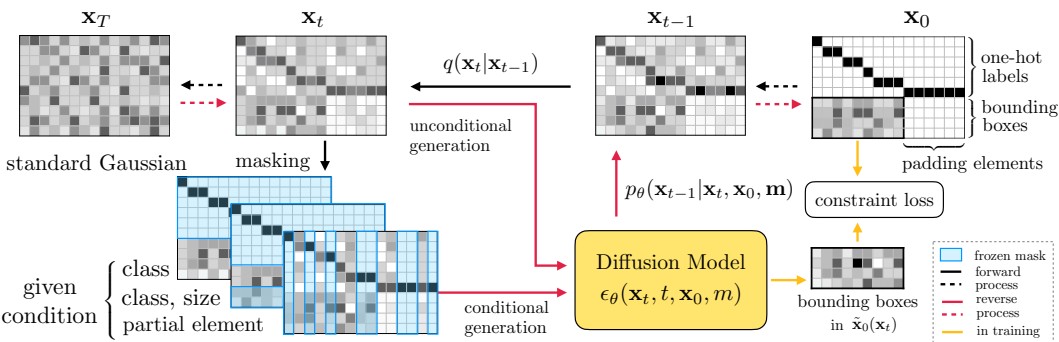

Figure 2: Overview of the layout generation model: The figure shows a layout generation of up to 15 elements from 5 classes. The layout is padded for consistency using padding elements, represented by an extra class in the 6-dimensional one-hot vector. The bounding box uses a 4-dimensional continuous vector. Dashed lines represent multi-step processes and solid lines represent a single step. The forward process corrupt data with Gaussian noise, while the reverse process trains a neural network to denoise the noisy latent $\mathbf{x}_t$ and its three augmentations using condition masks. Predicted bounding boxes is used to compute constraint loss against real ones.

Specifically, by rewriting the closed-form sampling distribution for the forward process, we have $\mathbf{x}_t = \sqrt{\bar{\alpha}_t}\mathbf{x}_0 + \sqrt{1 - \bar{\alpha}_t}\boldsymbol{\epsilon}$, $\boldsymbol{\epsilon} \sim \mathcal{N}(0, \mathbf{I})$. This equation yield a way to predict $\mathbf{x}_0$ at each time step using the predicted noise $\boldsymbol{\epsilon}_\theta$ as:

$$\tilde{\mathbf{x}}_0(\mathbf{x}_t) = (\mathbf{x}_t - \sqrt{1 - \bar{\alpha}_t} \cdot \boldsymbol{\epsilon}_\theta(\mathbf{x}_t, t))/\sqrt{\bar{\alpha}_t}. \tag{6}$$

The reconstruction loss is typically defined as the Mean Squared Error (MSE) between the prediction $\tilde{\mathbf{x}}_0$ and data $\mathbf{x}_0$. However, when we reframe the Layout generation as a continuous diffusion process, the results often exhibit poor global alignment and undesirable overlaps. This phenomenon occurs because the diffusion loss and the FID metric are not sensitive to minor variations in coordinates, yet they can cause tangible differences in visual quality. Since the contiuous search space for geometric attributes is much larger than in a discrete diffusion model, the diffusion loss holds many local minimums with similar FIDs and varying alignment scores. To address these issues, we propose to combine two additional aesthetic constraints in the reconstruction loss:

$$\mathcal{L}_{\text{rec}} = \text{MSE}(\tilde{\mathbf{x}}_0, \mathbf{x}_0) + \omega_t \cdot (\mathcal{C}_{\text{alg}}(\tilde{\mathbf{x}}_0(\mathbf{x}_t), \mathbf{x}_0) + \mathcal{C}_{\text{olp}}(\tilde{\mathbf{x}}_0(\mathbf{x}_t))), \tag{7}$$

where $\omega_t$ is a time-dependent weight to prevent convergence to local minimum. $\mathcal{C}_{\text{olp}}$ is the overlap constraint function, while $\mathcal{C}_{\text{alg}}$ can refer to either the local or global alignment constraint (represented as $\mathcal{C}_{\text{l-alg}}$, $\mathcal{C}_{\text{g-alg}}$), which will be defined below.

**Alignment constraint**   The alignment loss (Li et al., 2020) is a common metric to evaluate the alignment between layout elements for aesthetics assessment. While calculating this metric, bounding boxes are denoted by its six coordinates $\mathbf{b}_i = (b_i^{\text{L}}, b_i^{\text{XC}}, b_i^{\text{R}}, b_i^{\text{T}}, b_i^{\text{YC}}, b_i^{\text{B}})$ corresponds to six possible alignment types: (L), X-center (XC), right (R), top (T), Y-center (YC), and bottom (B) alignment. Since this loss function promotes an alignment pattern where each element is aligned with just one other element based on one type, we term it *local alignment* loss. For a set of $l$ bounding boxes $\mathbf{b}_1, \cdots, \mathbf{b}_l$ in a layout $\mathbf{x}$, the local alignment loss is defined as:

$$\mathcal{C}_{\text{l-alg}}(\mathbf{x}) = \sum_{i=1}^{l} \min \begin{pmatrix} g(\Delta b_i^{\text{L}}), & g(\Delta b_i^{\text{XC}}), & g(\Delta b_i^{\text{R}}) \\ g(\Delta b_i^{\text{T}}), & g(\Delta b_i^{\text{YC}}), & g(\Delta b_i^{\text{B}}) \end{pmatrix}, \tag{8}$$

where $g(x) = -log(1 - x)$, $\Delta b_i^* = \min_{\forall j \neq i} |b_i^* - b_j^*|$, $* \in \mathcal{A} = \{\text{L, XC, R, T, YC, B}\}$, and the minimum of the matrix above is defined as the smallest value among all its elements.

However, local alignment pattern is inconsistent with alignment observed in real-world graphic designs. For instance, in the context of website layouts, there exist elements that do not align with any others, while in publication layouts, certain elements might align with multiple others in multiple ways.

In order to encourage global alignment of elements, we propose a novel alignment loss function that encourages the model to learn global alignment pattern in the real data using an alignment mask. Specifically, for each of the six alignment types, we define the coordinate difference matrix for a layout $\mathbf{x}$ as $\mathbf{A}^*(\mathbf{x})_{i,j} = |b_i^* - b_j^*|$, $* \in \mathcal{A}$. Then the alignment loss function between a given prediction $\tilde{\mathbf{x}}$ at time $t$ and a real layout $\mathbf{x}$ is defined as:

$$\mathcal{C}_{\text{g-alg}}(\tilde{\mathbf{x}}, \mathbf{x}) = -\log\left(1 - \frac{1}{6}\sum_{* \in \mathcal{A}} \frac{\left\|\mathbf{A}^*(\tilde{\mathbf{x}}) \circ \mathbf{1}_{\mathbf{A}^*(\mathbf{x})=0}\right\|_1}{\left\|\mathbf{1}_{\mathbf{A}^*(\mathbf{x})=0}\right\|_1}\right), \tag{9}$$

where $\mathbf{1}_{\mathbf{A}^*(\mathbf{x})=0}$ is the binary ground truth alignment mask matrix selecting the zero entries of the coordinate difference matrix of the true layout, which represents the true alignment pattern in $\mathbf{x}$, and $\circ$ is the Hadamard product.

**Overlap constraint**  To prevent overlapping elements in the generated layout, $\tilde{\mathbf{x}}$, we apply a mean pairwise intersection over union loss function, characterized by the pairwise IoU matrix, $\mathbf{O}(\mathbf{x})_{ij} = (\mathbf{b}_i \cap \mathbf{b}_j)/(\mathbf{b}_i \cup \mathbf{b}_j)$. This function monotonically identifies similarities between sets but it has a continuous plane of stationary points, corresponding to the cases when a bounding box is fully contained within another. In such situations, infinitesimal adjustments to the bounding boxes do not cause the enclosed box to exceed the boundary of the larger box. To further penalize aforementioned instances, we define another matrix using the distance between centers of boxes, $\mathbf{D}(\mathbf{x})_{i,j} = e^{-\sqrt{(b_i^{\text{XC}} - b_j^{\text{XC}})^2 + (b_i^{\text{YC}} - b_j^{\text{YC}})^2}}$, which has value range from 0 to 1. Then we combine it with a mask matrix selecting overlapping pairs using the pairwise IoU matrix. The overlap constraint function is given as:

$$\mathcal{C}_{\text{olp}}(\tilde{\mathbf{x}}) = \text{mean}(\mathbf{O}(\tilde{\mathbf{x}}) + \mathbf{D}(\tilde{\mathbf{x}}) \circ \mathbf{1}_{\mathbf{O}(\tilde{\mathbf{x}})\neq 0}). \tag{10}$$

Minimizing the $\mathbf{D}(\mathbf{x})$ term push elements away from each other when one is entirely encompassed by another and for disjoint element pairs, the mask $\mathbf{1}_{\mathbf{x}\neq 0}$ deactivates the divisive affect.

**Time-dependent constraint weight**  In our observations, directly applying the alignment and overlap loss to noisy layout hinders the faithful reconstruction. This is due to the constraint function introducing numerous local minima within the parameter space as demonstrated in Figure B.2. To mitigate this, we employ a series of time-dependent constraint weight to enforce the constraint only for smaller time $t$ to finetune the misaligned coordinates in a less noisy prediction $\tilde{\mathbf{x}}_0$. We empirically choose $\omega_t = (1 - \bar{\alpha}_t)$ of a constant $\beta$ schedule as the constraint weight series. A detailed explanation and illustration is provided in Appendix B.1.

**Post-processing**  The model exhibited enhanced performance across FID, alignment, and Max-IoU metrics when trained with constraints. However, LACE may generate layouts with minor misalignment as the coordinate searching space is much larger than that of the discrete diffusion model. Fortunately, further refinement of its visual quality is straightforward and efficient. Guided by the strategies outlined in (Kikuchi et al., 2021), we employ constraint optimization in the post-processing phase, focusing on global alignment and, when suitable, the overlap constraint, targeting the geometric attributes directly. The application of global alignment demands an alignment mask, selecting coordinate pairs for minimization. In training, this mask is calculated using training data, ensuring the accurate learning of human-designed alignment patterns, as introduced in Eq. (9). Since there are no ground truth in the inference post-processing stage, we propose to use a threshold $\delta$ to identify nearly aligned entries in the coordinate difference matrix of the produced layout. Thus, the post-process forces elements to align with each other based on the pattern inferred from the raw output. It should be noted that an excessively large threshold will force more alignment, erasing the subtle structure of a layout and leading to coarse or even collapsing generation. A small threshold, on the other hand, might be ineffective in inducing modifications. Thus, the raw outputs must have a low coordinate difference in misaligned entries. In our experiment we set $\delta$ equal to $1/64$ of the scaled canvas range, as explained in Appendix B.2.

## 3  RELATED WORK

**Layout Generation**  Various automatic layout generation methods have emerged to aid the efficient creation of visually attractive and organized content across digital and print mediums, including websites, magazines, and mobile apps. Early works (O'Donovan et al., 2014; O'Donovan et al.,

2015) create layouts by optimizing energy-based objectives derived from design principles. With the advent of deep generative models, data-driven models LayoutGAN (Li et al., 2020) and LayoutVAE (Jyothi et al., 2019) are developed and enables more flexible conditional layout generation. Layout-GAN++ (Kikuchi et al., 2021) improves the performance of LayoutGAN for unconditional layout generation and proposes a CLG-LO framework to optimize a pre-trained LayoutGAN++ model to fulfill aesthetic constraints. LayoutTransformer (Gupta et al., 2021) and VTN (Arroyo et al., 2021) are autoregressive models based on transformer architectures, they improve the generation diversity and quality. However, the autoregressive mechanism hinders their application on conditional generation. BLT (Kong et al., 2022) introduces a bidirectional transformer encoder similar to masked language models to facilitate conditional generation with transformer models. However, the BLT model requires specifying the number of elements in advance. Thus, it can not solve completion tasks based on partial layout. LayoutFormer++ (Jiang et al., 2023) introduces a decoding strategy in its transformer encoder-decoder architecture to enable high-quality, conditional layout generation in an autoregressive manner. However, its backtracking mechanism may necessitate multiple reversions in the decoding process to rectify any invalid generations. Additionally, retraining is necessary for adapting to different conditional tasks. Zheng et al. (2019) incorporated image and text features into their layout generation model to achieve content-awareness.

**Diffusion-based layout generation** Recently, diffusion models have been adopted to develop unified models that adapt to various conditions and improve generation quality. PLay (Cheng et al., 2023) uses a latent diffusion model to generate layout conditioned on guidelines—lines align elements. LayoutDM (Inoue et al., 2023) and LDGM (Hui et al., 2023) build a unified model for various generation conditions without guidelines using the discrete diffusion (Austin et al., 2021) framework. They independently developed the same attribute-specific corruption strategy to restrict variables of different attributes to their respective sample spaces in the mask-and-replace forward process (Gu et al., 2022). In addition, LDGM employs discretized Gaussian noise to facilitate gradual coordinate changes. LayoutDiffusion (Zhang et al., 2023), adopting a similar design to LDGM, enhances visual quality of three generation tasks significantly by using a larger transformer backbone, particularly improving alignment and overlap metrics. In contrast, our method aims to enhance visual quality by applying constraint functions without scaling up the network architecture. Motivated by the superior performance of DDPM (Ho et al., 2020) in image generation, two recent studies have aimed to generate layouts in a continuous space. A diffusion-based model for unconditional generation was proposed by (He et al., 2023). Although it generate embedding vectors in continuous space, geometric attributes are quantized to discrete tokens, thus, is not differentiable. Another work (Chai et al., 2023) adopt DDPM for conditional generation that directly generate continuous geometric attributes (size and position) based on given categorical attributes. Because the noisy attributes share the same continuous sample space, DDPM-based models do not need the attribute-specific corruption strategy. Our model further innovates by directly generating geometric and categorical attributes in continuous space and utilizing differentiable aesthetic constraint functions to enhance generation quality. In addition, we train diffusion models using both masked and unmasked layouts similar to DiffMAE (Wei et al., 2023) to unify various tasks in a single model.

## 4 EXPERIMENTS

### 4.1 EXPERIMENT SETUP

**Datasets** We use two large-scale datasets for comparison. PubLayNet (Zhong et al., 2019) consists of 330K document layouts with five class element anotation retrived from articles from PubMed Centra™. Rico (Deka et al., 2017) consists of 72k user interfaces designs from 9.7k Android apps spanning 27 classes. Following LayoutDM (Inoue et al., 2023), for both datasets, we discard instance that has more than 25 elements, and pad the remaining layouts to a unified maximum length of 25. For Rico dataset, we use layouts that has only the most frequent 25 classes. The refined PubLayNet and Rico datasets were then split into training, validation, and test sets containing $315,757/16,619/11,142$ and $35,851/2,109/4,218$ samples respectively.

**Evaluation Metrics** To assess the quality of generation, we employ four computational metrics. (1) Fréchet Inception Distance (FID) (Heusel et al., 2017) compute the distance between the feature distributions of generated and real layout in the feature space of a pre-trained feature extraction

model. It evaluate both fidelity and diversity of generative models. We employ the same feature extraction model used in LayoutDM to compute FID. (2) Maximum Intersection-over-Union (Max.) is a metric for conditional generation. It compute the average IoU between optimally matched pairs of elements from two layouts that contain elements with identical category sets. (3) Alignment (Align.) and (4) Overlap is used for aesthetics assessment which compute the normalized alignment score. We adopt the implementation of all metrics as proposed in LayoutGAN++ (Kikuchi et al., 2021).

**Generation Tasks** We train a unified model for five generation tasks defined in previous works (Gupta et al., 2021; Jiang et al., 2022; Rahman et al., 2021) including unconditional generation (**U-Cond**), conditional generation based solely on class (**C→S+P**), conditional generation based on both class and size (**C+S→P**) of each element, **completion** with attributes given for a subset of elements, and **refinement** of a noisy layout. For completion task, following LayoutDM (Inoue et al., 2023), we randomly sample 0% to 20% of elements from real samples as the binary condition mask. In the completion task, we follow LayoutDM ((Inoue et al., 2023)), where we create a binary condition mask by randomly sampling between $0\%$ and $20\%$ of elements from real samples. In the refinement task, following RUITE ((Rahman et al., 2021)), we introduce minor Gaussian noise to the size and position of each element, characterized by a mean of $0$ and a variance of $0.01$, to simulate a perturbed layout. For more implementation details, please see Appendix B.

## 4.2 QUANTITATIVE RESULTS

We benchmark our method, LACE, against several state-of-the-art (SOTA) models on five generation tasks using the PubLayNet and Rico datasets. These models are categorized into traditional and diffusion-based models. Traditional models are further split into task-specific models, which include

| **PubLayNet** | Task | C→S+P | | C+S→P | | Completion | | U-Cond | |
|---|---|---|---|---|---|---|---|---|---|
| Model | Metric | FID↓ | Max.↑ | FID↓ | Max.↑ | FID↓ | Max.↑ | FID↓ | Align.↓ |
| *Task-specific models* | | | | | | | | | |
| LayoutVAE | | 26.0 | 0.316 | 27.5 | 0.315 | - | - | - | - |
| NDN-none | | 61.1 | 0.162 | 69.4 | 0.222 | - | - | - | - |
| LayoutGAN++ | | 24.0 | 0.263 | 9.94 | 0.342 | - | - | - | - |
| *Task-agnostic models* | | | | | | | | | |
| LayoutTrans | | 14.1 | 0.272 | 16.9 | 0.320 | 8.36 | 0.451 | 13.9 | 0.127 |
| BLT | | 72.1 | 0.215 | 5.10 | 0.387 | 131 | 0.345 | 116 | 0.153 |
| BART | | 9.36 | 0.320 | 5.88 | 0.375 | 9.58 | 0.446 | 16.6 | 0.116 |
| MaskGIT | | 17.2 | 0.319 | 5.86 | 0.380 | 19.7 | 0.484 | 27.1 | 0.101 |
| *Diffusion-based models* | | | | | | | | | |
| VQDiffusion | | 10.3 | 0.319 | 7.13 | 0.374 | 11.1 | 0.373 | 15.4 | 0.193 |
| LayoutDM | | 7.95 | 0.310 | 4.25 | 0.381 | 7.65 | 0.377 | 13.9 | 0.195 |
| LACE (local) | | 4.88 | 0.331 | 2.80 | 0.437 | 5.86 | 0.401 | 8.45 | 0.141 |
| LACE (global) | | 5.14 | 0.383 | 3.07 | 0.463 | 6.03 | 0.396 | 8.35 | 0.185 |
| LACE (local) w/ post | | 4.63 | 0.390 | 2.69 | 0.462 | 5.90 | 0.399 | 8.47 | 0.032 |
| LACE (global) w/ post | | 4.56 | 0.388 | 2.53 | 0.463 | 5.63 | 0.394 | 7.43 | 0.074 |
| Validation data | | 6.25 | 0.438 | 6.25 | 0.438 | 6.25 | 0.438 | 6.25 | 0.021 |
| **Rico** | Task | C→S+P | | C+S→P | | Completion | | U-Cond | |
| Model | Metric | FID↓ | Max.↑ | FID↓ | Max.↑ | FID↓ | Max.↑ | FID↓ | Align.↓ |
| *Task-specific models* | | | | | | | | | |
| LayoutVAE | | 33.3 | 0.249 | 30.6 | 0.283 | - | - | - | - |
| NDN-none | | 28.4 | 0.158 | 62.8 | 0.219 | - | - | - | - |
| LayoutGAN++ | | 6.84 | 0.267 | 6.22 | 0.348 | - | - | - | - |
| *Task-agnostic models* | | | | | | | | | |
| LayoutTrans | | 5.57 | 0.223 | 3.73 | 0.323 | 3.71 | 0.537 | 7.63 | 0.068 |
| BLT | | 17.4 | 0.202 | 4.48 | 0.340 | 117 | 0.471 | 88.2 | 1.030 |
| BART | | 3.97 | 0.253 | 3.18 | 0.334 | 8.87 | 0.527 | 11.9 | 0.090 |
| MaskGIT | | 26.1 | 0.262 | 8.05 | 0.320 | 33.5 | 0.533 | 52.1 | 0.015 |
| *Diffusion-based models* | | | | | | | | | |
| VQDiffusion | | 4.34 | 0.252 | 3.21 | 0.331 | 11.0 | 0.541 | 7.46 | 0.178 |
| LayoutDM | | 3.55 | 0.277 | 2.22 | 0.392 | 9.00 | 0.576 | 6.65 | 0.162 |
| LACE (local) | | 3.31 | 0.336 | 2.66 | 0.418 | 5.09 | 0.518 | 4.71 | 0.107 |
| LACE (global) | | 3.24 | 0.340 | 2.87 | 0.418 | 4.45 | 0.527 | 4.63 | 0.117 |
| LACE (local) w/ post | | 2.88 | 0.347 | 2.17 | 0.430 | 3.82 | 0.539 | 3.99 | 0.031 |
| LACE (global) w/ post | | 3.24 | 0.344 | 2.16 | 0.428 | 4.30 | 0.540 | 4.51 | 0.035 |
| Validation data | | 1.85 | 0.691 | 1.85 | 0.691 | 1.85 | 0.691 | 1.85 | 0.109 |

Table 1: Quantitative results on PubLayNet and Rico for four generation tasks. The top two results are highlighted with deep and light red shades, respectively.

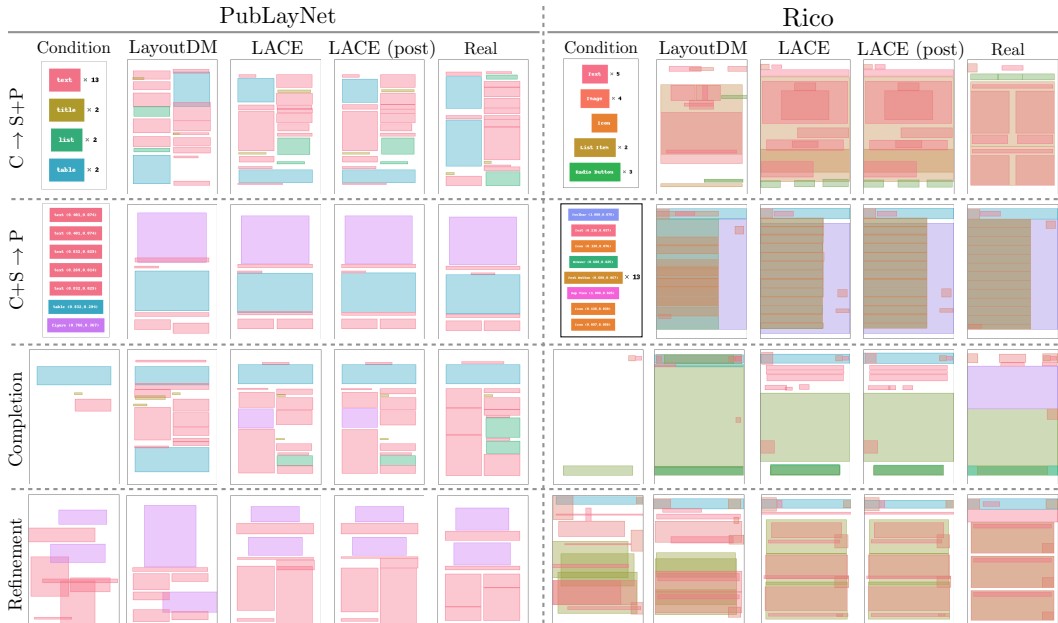

Figure 3: Qualitative comparison between LACE and LayoutDM in conditional generation tasks.

LayoutVAE (Jyothi et al., 2019), NDN-none (Lee et al., 2020), LayoutGAN++ (Kikuchi et al., 2021), and RUITE (Rahman et al., 2021), and task-agnostic models like LayoutTrans (Gupta et al., 2021), BLT (Kong et al., 2022), and MaskGIT (Chang et al., 2022). On the other hand, the diffusion-based models considered are VQDiffusion (Gu et al., 2022) and LayoutDM (Inoue et al., 2023). For evaluating unconditional generation, we utilize FID and Alignment metrics. Meanwhile, for refinement and conditional generation based on partial attributes, we apply FID and MaxIoU. Since the overlap patten is prevalent in the Rico dataset, we only apply the overlap constraint during training on the PubLayNet dataset. Therefore the overlap metric is only used in experiments on the PubLayNet dataset. Furthermore, results from models using both local and global alignment constraints are reported in the benchmark experiments. In the post-processing stage, we optimize the global alignment using the $\delta$-alignment mask introduced in section 2.3.

**Conditional and Unconditional Generation** We demonstrate that LACE, even without post-processing, achieves state-of-the-art (SOTA) performance in four tasks including unconditional and conditional generation given partial attributes. LACE surpasses other models that utilize discrete diffusion in benchmark experiments conducted on the Rico and PubLayNet datasets. The results for LACE, along with nine baseline methods, are presented in Table 1 and Appendix A. We observed Both LACE versions, using local and global alignment constraints, show similar performance. LACE (global) exhibited superior performance on the PubLayNet dataset, possibly due to the greater prevalence of alignment patterns in PubLayNet. The results of LACE with post-processing show a significant improvement in alignment, while either maintaining or slightly enhancing the FID. This is attributed to the adjustments that fine-tune the alignment without substantially altering the global organization of elements. Figure.3 shows the qualitative comparison results between LACE and LayoutDM for conditional generation. Additional qualitative results are in the Appendix C.

**Refinement** Continuous diffusion using Gaussian noise is naturally an expert at refining noisy layouts because noisy layouts can be viewed as noisy states during the forward process. And the continuous diffusion model is trained to reverse the corruption. Since it is only beneficial to refine a layout that is clear enough to show the relative position of elements, we assume the noise level of the input matches the noise states at time step $\tau$, where the time-dependent constraint weight starts to encourage the importance of the constraint functions ($\omega_t = 0.1$). Thus, we use a fixed LACE model to refine a noisy layout, treat the input as a noisy state at time $\tau$, and start a partial reverse process to refine it. The results shown in Table 2 demonstrate LACE outperforms competing models, achieving FID and MaxIoU scores comparable to that of LACE with post-processing.

| Dataset | Rico | | PubLayNet | |
|---|---|---|---|---|
| Model | FID↓ | Max↑ | FID↓ | Max↑ |
| Task-specific models | | | | |
| RUITE (Rahman et al., 2021) | 263.23 | 0.421 | 6.39 | 0.415 |
| Task-agnostic models | | | | |
| Noisy input | 134 | 0.213 | 130 | 0.242 |
| LayoutDM | 2.77 | 0.370 | 4.25 | 0.381 |
| LACE (local) | 1.79 | 0.485 | 1.65 | 0.491 |
| Post-processed | | | | |
| LACE (local) | 1.76 | 0.484 | 1.80 | 0.500 |
| Validation data | 6.25 | 0.438 | 6.25 | 0.438 |

Table 2: Results in the refinement task on PubLayNet and Rico datasets.

## 4.3 ABLATION STUDY

We conducted an ablation study on the PubLayNet dataset to evaluate the role of aesthetic constraints and masked input during training. We trained a LACE model without aesthetic constraints and three diffusion models with constraints for unconditional generation and two conditional tasks (C→S+P, C+S→P). Using the unconditional model, we also applied the in-painting technique (Lugmayr et al., 2022) for completion. We only include LACE using local alignment constraint (noted as LACE w/ $\mathcal{C}$), in the ablation study, as LACE with both constraints exhibit similar performance. Table 3 displays the results. The result of task-specific models is comparable to LACE, which proves the effectiveness of our task unification method. LACE achieves a much lower alignment score and slightly reduced FID compared to LACE without aesthetic constraints, suggesting the effectiveness of constraint optimization. In addition, the post-process notably improves the alignment in unconditional generation without sacrificing the FID score. However, the post-process is not as effective for LACE without constraints. A possible explanation is the coordinate difference in the raw output is too large for the threshold to detect, resulting in a smaller number of coordinate differences being minimized. We include detailed alignment and overlap results in Appendix A.

| Task | Metric | Task-specific w/ $\mathcal{C}$ | LACE w/o $\mathcal{C}$ | LACE | LACE w/o $\mathcal{C}$ w/ post | LACE w/ post |
|---|---|---|---|---|---|---|
| U-Cond | FID↓ | 6.76 | 8.70 | 8.45 | 10.31 | 8.47 |
| | Align↓ | 0.128 | 0.238 | 0.141 | 0.215 | 0.032 |
| C→S+P | FID↓ | 6.12 | 5.08 | 4.88 | 4.76 | 4.63 |
| | Max↑ | 0.386 | 0.383 | 0.332 | 0.389 | 0.390 |
| C+S→P | FID↓ | 2.25 | 3.21 | 2.80 | 2.57 | 2.69 |
| | Max↑ | 0.478 | 0.460 | 0.437 | 0.461 | 0.462 |
| Completion | FID↓ | 7.73* | 6.42 | 5.86 | 6.01 | 5.90 |
| | Max↑ | 0.399 * | 0.403 | 0.401 | 0.385 | 0.399 |

Table 3: Results of the ablation study conducted on the PubLayNet dataset. The asterisk (*) denotes in-painting results using the unconditional model. Symbol $\mathcal{C}$ represents 'constraints', while 'post' refers to post-processing. The top two results are highlighted in deep and light red shades.

## 5 CONCLUSIONS

We introduced LACE, a diffusion a unified model for controllable layout generation in a continuous state space. Using the diffusion framework with alignment and overlap loss constraints, LACE outperforms previous state-of-the-art baselines in both FID and visual quality. Efficient post-processing can further improve generation quality regarding alignment and overlap, without sacrificing FID. While LACE demonstrates advancements in layout generation, it has several limitations. First, it restricts layout elements to box shapes, limiting representation flexibility. Additionally, it lacks background and content awareness, which could be crucial for more context-driven designs. The model also handles only a limited number of elements and relies on a closed label set. These constraints may restrict its applicability in complex, varied design scenarios. For future work, adopting arbitrary shapes can better mirror real-world graphic design scenarios as most existing work relies on rectangular boxes for element representation. In addition, using images or text as conditions to generate a more relevant layout for the content should improve the performance of downstream tasks, such as layout-guided image generation and automated web design.

## ACKNOWLEDGEMENT

This work is partially supported by NSF AI Institute-2229873, NSF RI-2223292, an Amazon research award, and an Adobe gift fund. Any opinions, findings and conclusions or recommendations expressed in this material are those of the author(s) and do not necessarily reflect the views of the National Science Foundation, the Institute of Education Sciences, or the U.S. Department of Education.

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

# A  ADDITIONAL ABLATION RESULTS

We compared LACE with LayoutDM and LayoutGAN++ in terms of overlap and alignment. LayoutGAN++ is a task-specific model that also uses constraints in training and post-processing. We directly adopt their results in the original paper. In addition, we apply post-processing to LayoutDM to demonstrate the advantage in using constraint during training for diffusion model. Results are shown in Table A.1 and Table A.2. Both LayoutDM and LayoutGAN++ show a notable FID increase after post-processing. In contrast, LACE maintains a stable FID and achieves lower overlap and alignment scores before post-processing, which are further improved afterward.

| Task | | C→S+P | | |
|---|---|---|---|---|
| Model | Metric | FID↓ | Align↓ | Overlap.↓ |
| *Task-specific models* | | | | |
| NDN-none | | 61.1 | 0.350 | 16.5 |
| LayoutGAN++ | | 24.0 | 0.190 | 22.80 |
| LayoutGAN++ w/ $\mathcal{C}$ | | 22.3 | 0.160 | 14.27 |
| LayoutGAN++ w/ $\mathcal{C}$ & post | | 26.2 | 0.160 | 1.18 |
| *Diffusion-based models* | | | | |
| LayoutDM | | 7.95 | 0.106 | 16.43 |
| LayoutDM w/ post | | 15.2 | 0.083 | 6.076 |
| LACE w/o $\mathcal{C}$ | | 6.12 | 0.054 | 1.636 |
| LACE (local) | | 4.88 | 0.043 | 1.638 |
| LACE (global) | | 5.14 | 0.046 | 1.791 |
| LACE (local) w/ post | | 4.63 | 0.010 | 1.211 |
| LACE (global) w/ post | | 4.56 | 0.009 | 0.906 |
| Validation data | | 6.25 | 0.021 | 0.117 |

Table A.1: FID, overlap and alignment results in the C→S+P task on the PubLayNet dataset.

Additionally, LACE, even without post-processing, outperforms LayoutDM with post-processing in alignment and overlap scores. This serves as proof of the effectiveness of the constraint loss in our approach.

| Task | | C+S→P | | Completion | | U-Cond | |
|---|---|---|---|---|---|---|---|
| Model | Metric | Align↓ | Overlap.↓ | Align↓ | Overlap.↓ | Align↓ | Overlap.↓ |
| *Diffusion-based models* | | | | | | | |
| LayoutDM | | 0.119 | 18.91 | 0.107 | 15.04 | 0.195 | 13.43 |
| LayoutDM w/ post | | 0.117 | 6.506 | 0.073 | 5.220 | 0.200 | 4.641 |
| LACE w/o $\mathcal{C}$ | | 0.065 | 3.062 | 0.054 | 3.223 | 0.238 | 7.533 |
| LACE (local) | | 0.061 | 3.309 | 0.040 | 2.772 | 0.141 | 3.615 |
| LACE (global) | | 0.061 | 3.439 | 0.042 | 3.056 | 0.185 | 4.140 |
| LACE (local) w/ post | | 0.016 | 1.400 | 0.014 | 1.723 | 0.032 | 0.586 |
| LACE (global) w/ post | | 0.017 | 1.363 | 0.017 | 1.573 | 0.074 | 0.768 |
| Validation data | | 0.021 | 0.117 | 0.021 | 0.117 | 0.021 | 0.117 |

Table A.2: Overlap and alignment results on PubLayNet for three generation tasks.

# B  IMPLEMENTATION SPECIFICATIONS

## B.1  TIME-DEPENDENT CONSTRAINT WEIGHT

The time-dependent constraint weight is critical for effective model convergence and output quality. Without this weight, the model struggles to converge, leading to a high Fréchet Inception Distance (FID) score, typically remaining above 100, which indicates poor layout quality. We choose $\omega_t = (1 - \bar{\alpha}_t)$ of a constant $\beta$ schedule as the constraint weight series. The $\beta$ schedule is set empirically such that the weight activates the constraint only when t is small when the corruption process has not introduced too much overlap, as demonstrated in Figure B.1. Thus, in the reverse process, the coarse structure of the layout has emerged. Figure B.2 demonstrates the local minimum induced by the constraint functions at noisy steps, hindering convergence.

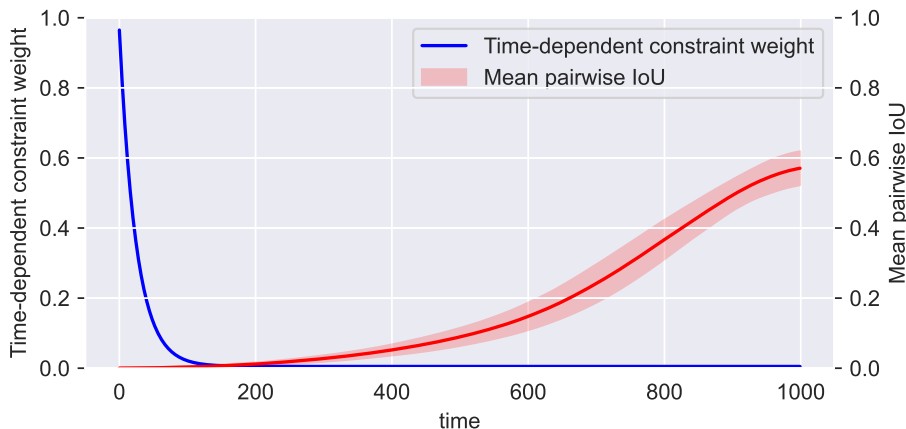

Figure B.1: Time-dependent constraint weight and Mean Pairwise IoU in the forward process.

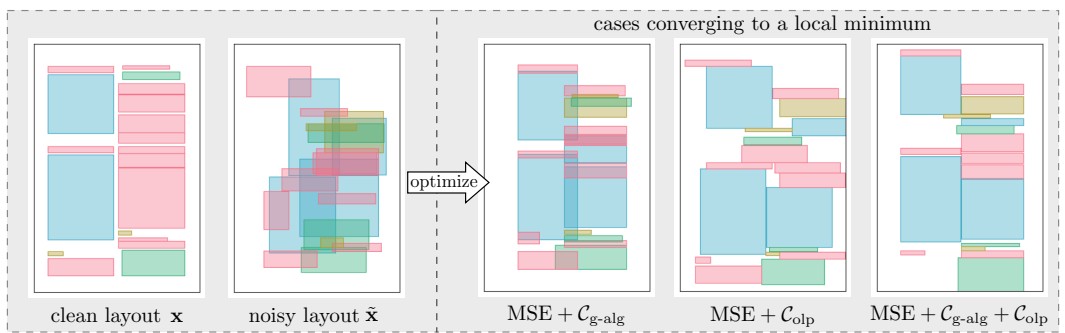

Figure B.2: Examples of convergence to local minimum with alignment and overlap constraints

## B.2 POST-PROCESSING THRESHOLD

We use the global alignment and overlap (only on PubLayNet) constraints to optimize the raw output of LACE. Since there is no target layout to compute the ground truth alignment mask matrix in Eq. (9), we use a threshold value to compute an alignment mask matrix using the coordinate difference matrix of the generated layout. To determine the threshold for enhanced visual quality post-processing, we first scaled the normalized canvas according to its width/height ratio. We then tested various threshold values, including 1/16, 1/32, 1/64, 1/128, and 1/256. The optimal threshold was empirically determined to be 1/64 of the scaled normalized canvas size. This setting aligns with real dataset observations, where only 0.5% of unaligned coordinate pairs have a smaller difference.

## B.3 MODEL ARCHITECTURE

As illustrated in Figure B.3, we adopt a transformer architecture that is implemented in the source code of LayoutDM Inoue et al. (2023) to predict the noise term in Eq. (5). We also choose similar hyper-parameter settings for a fair comparison: 4 layers, 16 attention heads, 2048 hidden dimension in the FNN (feed-forward networks), and the embedding dimension is 1024 for PubLayNet and 512 for Rico. In addition, we add two FNNs to encode and decode element vectors. Time embeddings are injected by an modified adaptive layer normalization Dumoulin et al. (2017). Specifically, the layer normalization is:

$$\mathbf{y} = (\mathbf{1} + f_\gamma(\mathbf{v}_t)) \odot \left( \frac{\mathbf{x} - \boldsymbol{\mu}}{\boldsymbol{\sigma}} \right) + f_\beta(\mathbf{v}_t), \tag{B.1}$$

where $\mathbf{x}$, $\mathbf{y}$ are the input and output of the normalization function, $f_\gamma$, $f_\beta$ are FFN that encode the time-dependent scale and shift, $\boldsymbol{\mu}$, $\boldsymbol{\sigma}$ are $\mathbf{x}$'s mean and standard deviation, $\mathbf{v}_t$ is the time embedding, $\mathbf{1}$ is a vector of all ones represents a residue connection. $\odot$ is the Hadamard product.

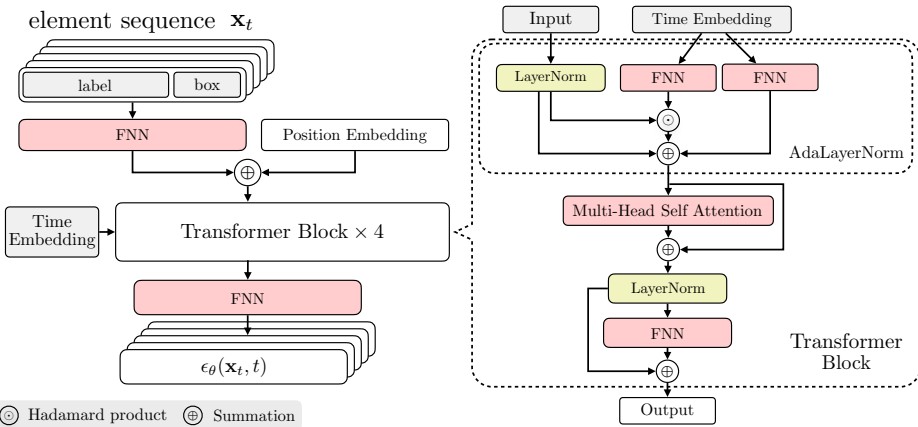

Figure B.3: Neural network architecture for the layout generation diffusion models. The the network takes sequence of layout elements the time variables as input and output the predicted noise. The pink blocks in the figure represent the trainable network components, gray blocks represent input tensors.

## B.4 TRAINING DETAILS

We train the model using the Adam optimizer. The batch size is 256. We used a learning rate schedule that included a warmup phase followed by a half-cycle cosine decay. The initial learning rate is set to 0.001. Training is divided into two phases: initially, the model is trained without constraints ($\omega_t = 0$) until convergence is observed in the FID score. In the second phase, constraints are added to the total loss, and training continues until convergence is achieved in both alignment and FID scores. For the Rico dataset, the overlap constraint is excluded due to prevalent overlap patterns in real data. However, in the PubLayNet dataset training, the overlap constraint is applied to prevent undesirable overlaps in publication layouts. The diffusion model employs a total of 1000 forward steps. For efficient generation, we use DDIM sampling with 100 steps.

## C QUALITATIVE COMPARISON

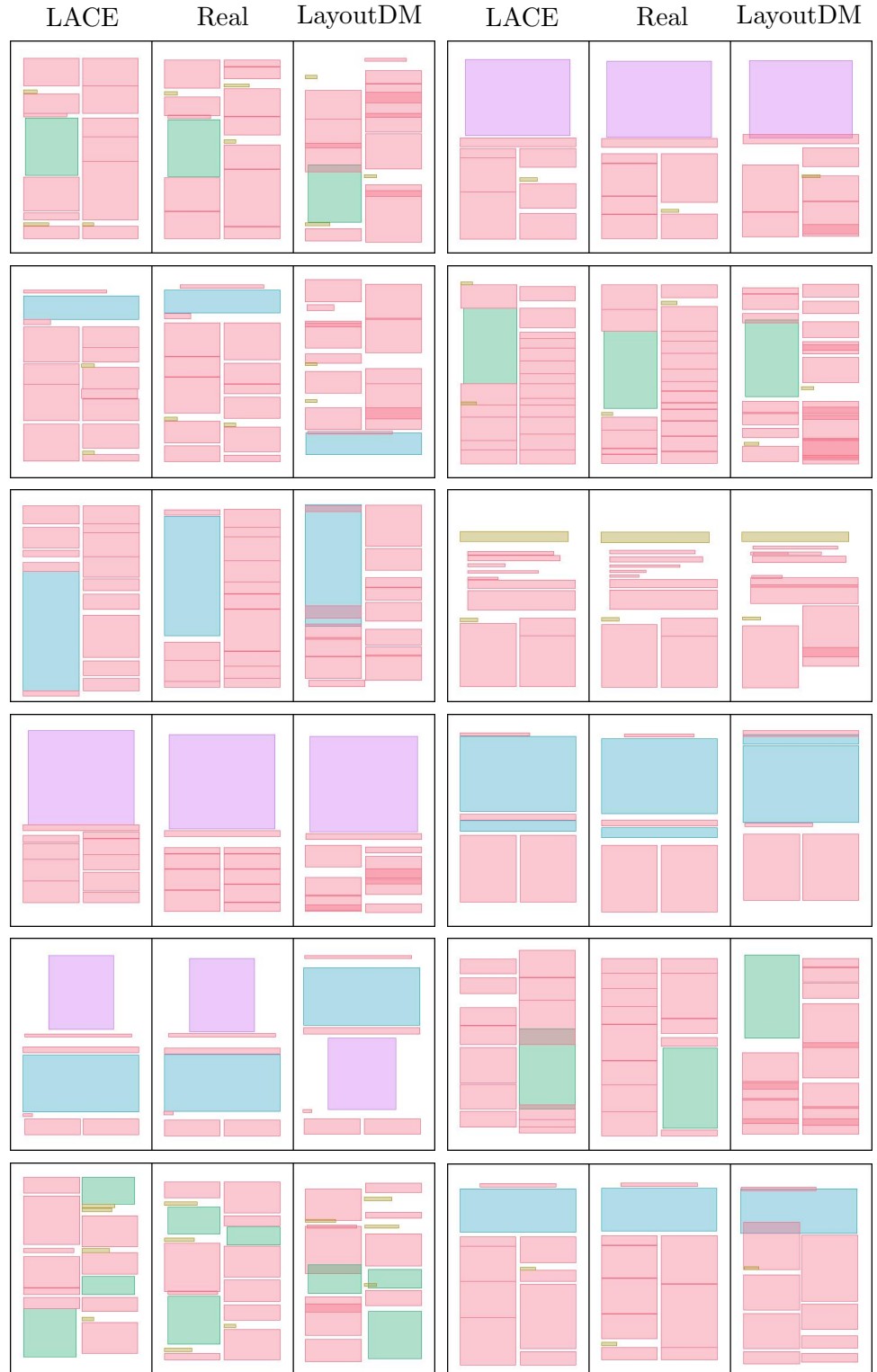

Figure C.1: Qualitative comparison between LACE w/ post-processing (left), real (middle), and LayoutDM (right) in conditional generation tasks (C+S → P) on the PublayNet dataset.

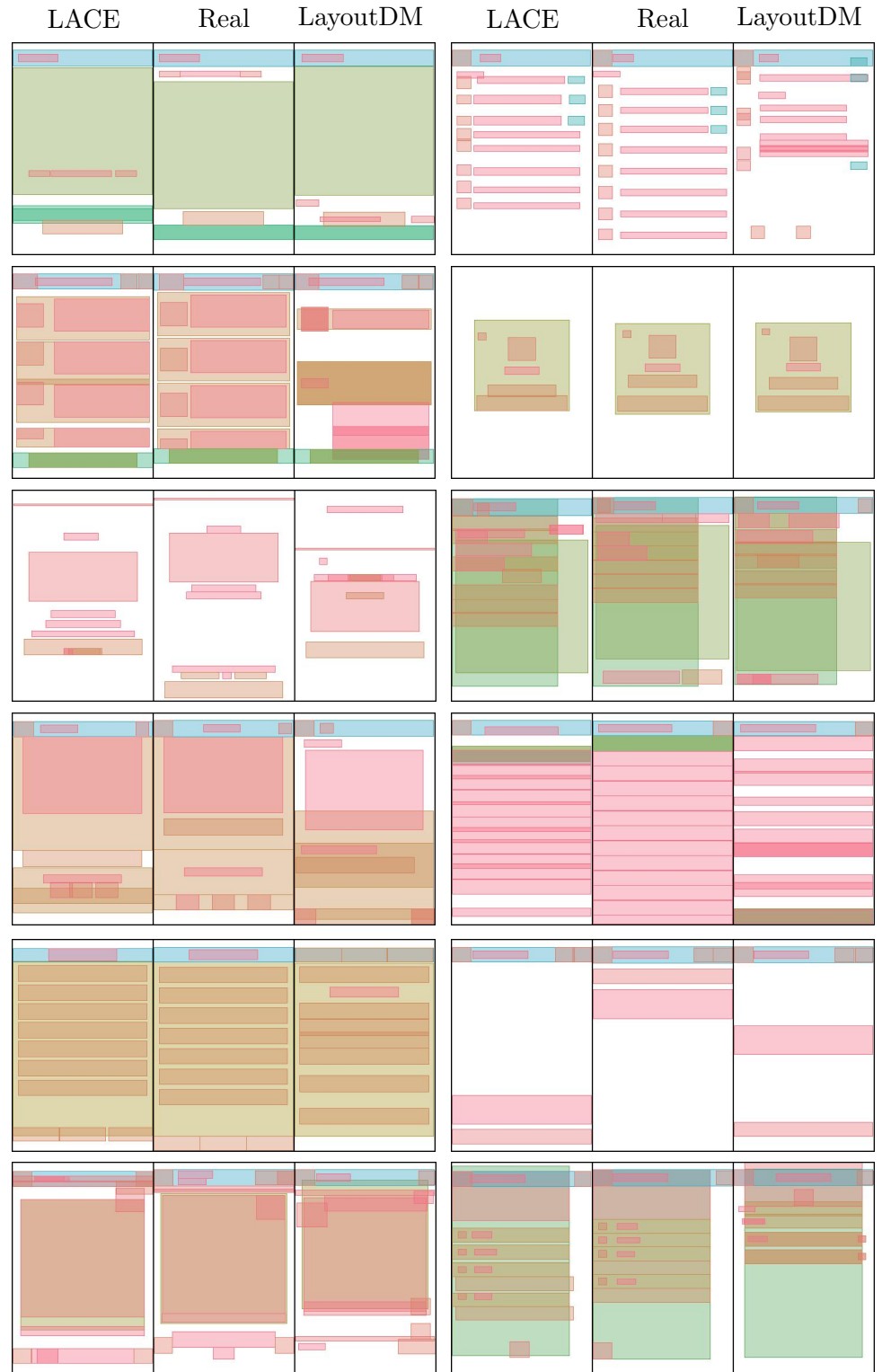

Figure C.2: Qualitative comparison between LACE w/ post-processing (left), real (middle), and LayoutDM (right) in conditional generation tasks (C+S → P) on the Rico dataset.

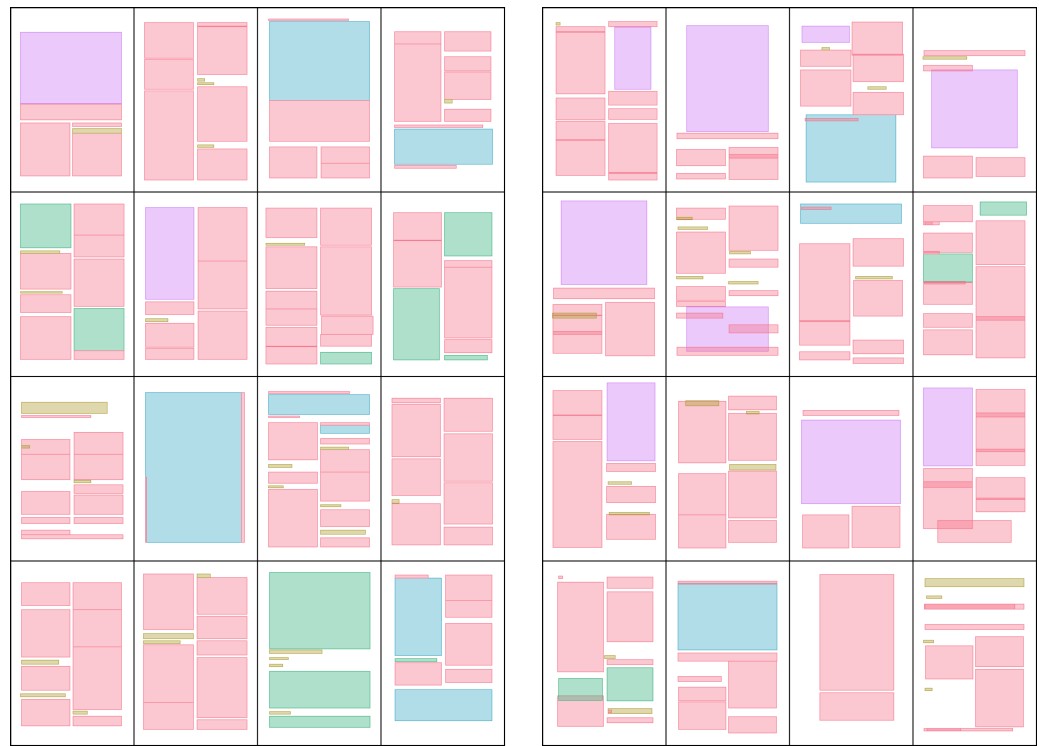

Figure C.3: Qualitative comparison between LACE w/ post-processing (left) and LayoutDM (right) in unconditional generation tasks on the PublayNet dataset.

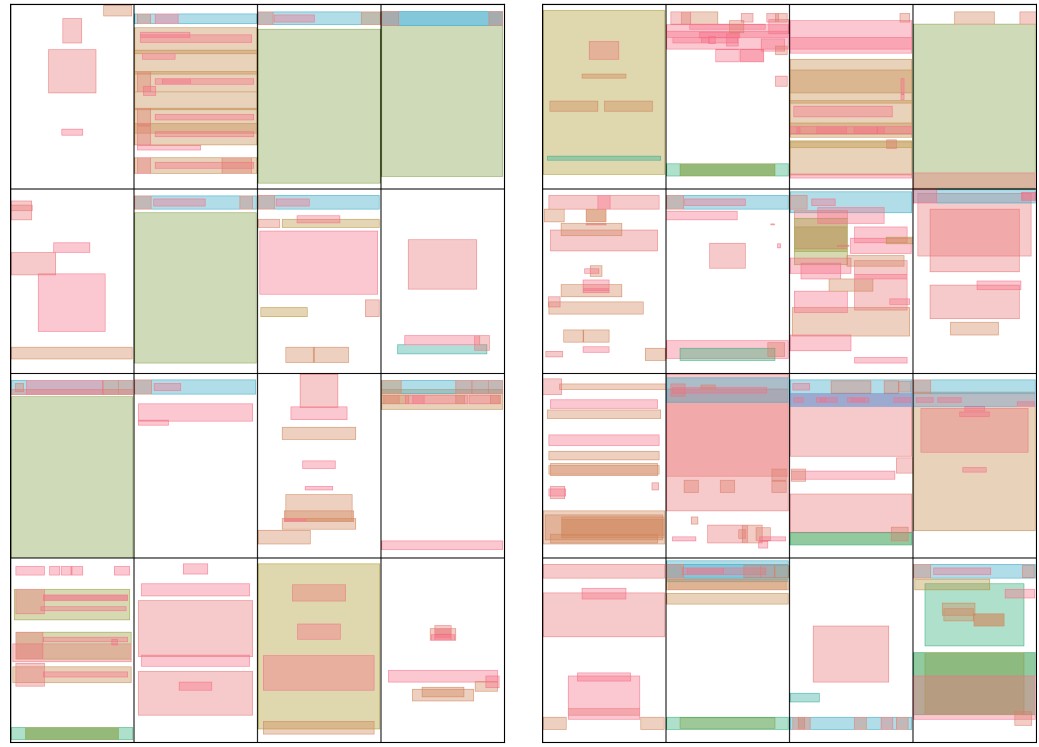

Figure C.4: Qualitative comparison between LACE w/ post-processing (left) and LayoutDM (right) in unconditional generation tasks on the Rico dataset.

