# OpenReview forum: "Towards Aligned Layout Generation via Diffusion Model with Aesthetic Constraints"
_ICLR.cc/2024/Conference — ICLR 2024 poster_

### Official Review · Reviewer_w7ws · 2023-10-28

**Soundness:** 3 good
**Presentation:** 4 excellent
**Contribution:** 3 good
**Rating:** 6
**Confidence:** 4

**Summary:**

This paper presents a novel unified continuous diffusion model, namely LAyout Constriant Diffusion modEl (LACE) for layout generation. To address the aesthetic issue of the diffusion model, the authors propose two differential constraints, including alignment constraint and overlap constraint, to address element arrangement and overlap issues. The experiments show that the proposed LACE makes a good performance on PubLayNet and Rico datasets.

**Strengths:**

* The paper is interesting, well-written and original, and my impression is positive.
* The proposed framework for various generation tasks is smart and effective.
* The quantitative experiment results show the superiority of the proposed method over existing approaches.

**Weaknesses:**

* I would appreciate it if the authors give more details of the proposed framework given different conditions.
* The ablation study can be extended to illustrate the contribution of two constraints. For example, Figure 3 only shows the visualization result of global alignment and overlap constraints. I would recommend studying the effectiveness of all sub-constraints, e.g., local and global alignment, quantitatively and qualitatively.
* The limitations of the proposed method should be discussed. This can give good guidance for the application in real-world situations.

**Questions:**

Please refer to the weakness part.

---

> ### Author Response · Authors · 2023-11-20
>
> W1: I would appreciate it if the authors give more details of the proposed framework given different conditions.
>
> A: Thank you for your feedback. We have added detailed information about our model architecture, hyper-parameter settings, and training details in Appendix B.3. In short, our model is a four-layer transformer with a modified adaptive layer normalization for time variable injection. This architecture is adopted from the source code of LayoutDM. We added two FNNs to encode and decode element vectors. The model size remains comparable. We use only one set of parameters for different conditional tasks. Known entries are fixed throughout the generation process using a binary mask as described at the end of section 2.2. For refinement task, we treat given noisy layout as a noisy latent states at a certain time step and finish the generation process as described at the end of section 4.2.
>
> W2: The ablation study can be extended to illustrate the contribution of two constraints. For example, Figure 3 only shows the visualization result of global alignment and overlap constraints. I would recommend studying the effectiveness of all sub-constraints, e.g., local and global alignment, quantitatively and qualitatively.
>
> A: Thank you for your suggestion. We have added more results in our ablation study and Appendix A. We observe that global alignment constraint performs comparably to local alignment. However, in theory, it should be less effective because local alignment constraints directly minimize the alignment metric. In addition, the global constraint is needed in post-processing for better visual quality.
>
> W3: The limitations of the proposed method should be discussed. This can give good guidance for the application in real-world situations.
>
> A: Thank you for your suggestion. We have added the limitation in the revised conclusion section. The primary limitations include (some also apply to related methods): 1. It generates layouts as boxes, not segmentations. 2. the model is not background-aware or content-aware. 3. The maximum number of elements is limited. 4. It relies on a closed-vocabulary label set.
>
> Despite these limitations, the method shows promise in applications like layout-guided image generation, realistic scene text generation through TextDiffuser, and automated poster design, offering valuable contributions.

---

### Official Review · Reviewer_5Gyc · 2023-10-30

**Soundness:** 2 fair
**Presentation:** 3 good
**Contribution:** 2 fair
**Rating:** 6
**Confidence:** 4

**Summary:**

This paper focuses on the controllable layout generation tasks, and formulates them as conditional generation processes in continuous space for aesthetic quality optimization. Two aesthetic constraint losses are proposed for global alignment and minimizing overlap in the layout during the training and post-processing stages. The experimental results show that the proposed method achieves the state-of-the-art performances on several benchmarks.

**Strengths:**

Compared to existing works that use discrete diffusion models, the proposed method based on continuous diffusion models can incorporate continuous aesthetic constraints in training. A new alignment loss is proposed to encourage the global alignment of elements. The proposed model can handle multiple generation tasks without retraining.

**Weaknesses:**

1. The key idea of this paper is to enable constraint optimization by formulating layout generation as conditional generation processes in continuous space. However, a post-processing step is still conducted. I was wondering about the advantage of the proposed method compared to those that directly apply the post-processing algorithm to the discrete diffusion models.
2. Several layout generation related works [1-3] are neither cited nor discussed in the paper.
3. For the overlap constraint, the current loss pushes elements away from each other. However, in some cases, designers would intentionally use overlap to make the top element look closer to the viewer. For example, text elements are often located above the image elements in banner ads. The usefulness of such a constraint would be questionable by directly applying it to all elements in graphic designs.
4. Why the evaluation metrics do not consider overlap, while the aesthetic constraints contain the overlap in layout?

[1] Zheng, Xinru, et al. "Content-aware generative modeling of graphic design layouts." ACM TOG, 2019.
[2] Zhang, Junyi, et al. "LayoutDiffusion: Improving Graphic Layout Generation by Discrete Diffusion Probabilistic Models." ICCV 2023.
[3] Jiang, Zhaoyun, et al. "LayoutFormer++: Conditional Graphic Layout Generation via Constraint Serialization and Decoding Space Restriction." CVPR 2023.

**Questions:**

1. How to determine the time-dependent constraint weight and the threshold used in the post-processing stage?
2. Are there any qualitative results that demonstrate diversity of the generated layouts?

---

> ### Author Response · Authors · 2023-11-20
>
> W1: The key idea of this paper is to enable constraint optimization by formulating layout generation as conditional generation processes in continuous space. However, a post-processing step is still conducted. I was wondering about the advantage of the proposed method compared to those that directly apply the post-processing algorithm to the discrete diffusion models.
>
> A: Thank you for your suggestion. Appendix A includes the required results and discussion, comparing our model with LayoutDM with post-processing. In addition, we include results of LayoutGAN++, which also uses constraints in training and applies post-processing.
>
> We observe a noticeable FID increase after post-processing for the two competing methods. In contrast, LACE without post-processing outperforms LayoutDM with post-processing. Its performance further increases after post-processing while maintaining a stable FID. These differences are due to the challenge in post-processing, as discussed in section 2.3 of the main text, which is thresholding the misalignment in raw outputs. A small threshold for the alignment mask is insufficient to select severely misaligned pairs, while a large one damages the subtle layout structure, leading to a noticeable increase in FID. These additional results have further enhanced the quality of our paper.
>
> W2: Several layout generation related works [1-3] are neither cited nor discussed in the paper.
> [1] Zheng, Xinru, et al. "Content-aware generative modeling of graphic design layouts." ACM TOG, 2019. [2] Zhang, Junyi, et al. "LayoutDiffusion: Improving Graphic Layout Generation by Discrete Diffusion Probabilistic Models." ICCV 2023. [3] Jiang, Zhaoyun, et al. "LayoutFormer++: Conditional Graphic Layout Generation via Constraint Serialization and Decoding Space Restriction." CVPR 2023.
>
> A: Thank you for your valuable feedback and the suggestion to include these recently published methods. We discuss the relationship between these methods and LACE and introduce and cite these methods in the revised related work section.
>
> 1. LayoutDiffusion adopted a design similar to LDGM (Hui, Mude, et al. "Unifying Layout Generation with a Decoupled Diffusion Model." CVPR 2023) significantly enhanced the visual quality in three tasks by using a more effective transformer backbone, particularly improving alignment and overlap metrics. In contrast, our method aims to enhance visual quality by applying constraint functions without scaling up or complicating the network architecture.
> 2. LayoutFormer++ employs a novel decoding strategy for high-quality conditional generation. However, its backtracking mechanism can result in multiple rollbacks during the decoding process to address invalid outputs. Moreover, it requires retraining to adapt to various conditional tasks. In contrast, LACE aims to finish generation without rolling back and use one set of parameters to solve different tasks.
> 3. The Content-aware Method in [1] solves a more challenging problem by incorporating image and text features into their layout generation model to achieve content-awareness.
>
> Due to the notable differences in model size and architecture between the methods we included in our experiment and that in LayoutDiffusion and LayoutFormer++, a quantitative comparison would be unfair. Consequently, our discussion of these methods is limited to the related work section.

---

> ### Author Response · Authors · 2023-11-20
>
> W3: For the overlap constraint, the current loss pushes elements away from each other. However, in some cases, designers would intentionally use overlap to make the top element look closer to the viewer. For example, text elements are often located above the image elements in banner ads. The usefulness of such a constraint would be questionable by directly applying it to all elements in graphic designs.
>
> A: We apologize for any confusion regarding the overlap constraint. As noted in the global response, we acknowledge that the overlap constraint is unsuitable for training on the Rico dataset. Thus, we exclude it in the training of the model for Rico, which was noted in section 4.2. However, our method provided a way to add differentiable constraint functions in the diffusion training. The constraint could be replaced or removed as needed.
>
> W4: Why the evaluation metrics do not consider overlap, while the aesthetic constraints contain the overlap in layout?
>
> A: We apologize for the confusion. As mentioned in our global response, we have added the requested alignment and overlap results in Appendix A. We believe these additional results have further enhanced the quality of our paper.
>
> Q1: How to determine the time-dependent constraint weight and the threshold used in the post-processing stage?
>
> A: We have now included a detailed explanation with a supporting figure in Appendix B.1 to demonstrate our empirical approach for determining the time-dependent constraint weights. This weight is empirically set to the point, as demonstrated in the figure, where the corruption process has not introduced too much overlap. Thus, in the reverse process, the coarse structure of the layout has emerged.
>
> The post-processing threshold is empirically set to 1/64 of the scaled relative canvas size. This setting aligns with real dataset observations, where only ~0.5% of unaligned coordinate pairs has a smaller difference. We have included a more detailed explanation in Appendix B.2.
>
> Q2: Are there any qualitative results that demonstrate diversity of the generated layouts?
>
> A: In response to your query, we've included qualitative results in Appendix C, showcasing the diversity of layouts generated by our model, compared with real data and LayoutDM.

---

> ### Author Response · Authors · 2023-11-22
>
> We apologize for the oversight in our initial explanation about the similarity between LayoutDiffusion and prior works. Specifically, LayoutDiffusion incorporates discrete Gaussian noise in its design, differing from the approach in LayoutDM and aligning with the method proposed by LDGM. We have corrected this in our revised response to Weakness 2. Additionally, we have cited and discussed these papers in the revised manuscript. We hope this revision adequately addresses your concerns.

---

### Official Review · Reviewer_6Y7v · 2023-11-02

**Soundness:** 3 good
**Presentation:** 4 excellent
**Contribution:** 3 good
**Rating:** 8
**Confidence:** 3

**Summary:**

This paper introduces a novel diffusion model and training process for various conditional and unconditional layout generation tasks. Utilizing a continuous state-space design directly on the target attributes of elements in the layout, the proposed method incorporates both the standard diffusion training objective and novel aesthetic-based objectives (i.e., overlap, alignment) to improve the quality of generated layouts. These improvements lead to a new SoTA model across the majority of layout generation tasks in both the Rico and PubLayNet dataset.

**Strengths:**

- Novel formulation and diffusion model for directly using a continuous state-space for geometric and class properties, which could opens up a new avenue of research for layout generation by exploring diffusion in this space

- Unified framework for both unconditional generation and multiple conditional tasks

- Achieved SoTA in the majority of conditional tasks in both Rico and PubLayNet

- Novel formulation of Overlap and Alignment losses, and a time-dependent constraint schedule for effective optimization of aesthetic constraints during the diffusion process, which can independently benefit future research given the importance of these constraints in good layouts.

**Weaknesses:**

- *Incomplete details for reproduction*: It appears that no training details of the models are reported in the paper and the appendix. It would be extremely helpful for future research for the author(s) the report details such as model architecture, noise schedule, loss schedule, constraint weights, etc, given that this continuous state-space of diffusion for layout generation was previously unexplored.

- *Limited inclusion of qualitative results and human-rater experiments*: It would further improve the paper if the author(s) can provide more qualitative results that demonstrates the differences between the proposed method and existing SoTA, and preferably include a lightweight human-rating study given that the metrics might not fully reflect human preference for layout generation.

- *Missing common metrics*: While FID and MaxIoU is quite representative and established in the layout generation literature, it will be helpful if the author(s) can include alignment and overlap indices, which can further show the effect of the proposed aesthetic constraints.

**Questions:**

- Can the author(s) provide more details of model training (e.g., loss schedule, model architecture), given that the introduced method applied diffusion in a novel and complex state-space that is non-trivial?

- Will the code be open-sourced if this paper is accepted?

- Can you further discuss the importance of the time-dependent constraint weight (preferably empirically)? This appears to be a very important parameter in incorporating aesthetic constraints.

---

> ### Author Response · Authors · 2023-11-20
>
> W1 & Q1: Incomplete details for reproduction: It appears that no training details of the models are reported in the paper and the appendix. It would be extremely helpful for future research for the author(s) the report details such as model architecture, noise schedule, loss schedule, constraint weights, etc, given that this continuous state-space of diffusion for layout generation was previously unexplored. Can the author(s) provide more details of model training (e.g., loss schedule, model architecture), given that the introduced method applied diffusion in a novel and complex state-space that is non-trivial?
>
> A: Thank you for your feedback. We have added detailed information about our model architecture, hyper-parameter settings, and training details in Appendix B.3 and Appendix B.4. In short, our model is a four-layer transformer with a modified adaptive layer normalization for time variable injection. This architecture is adopted from the source code of LayoutDM. We added two FNNs to encode and decode element vectors. The model size remains comparable.
>
> W2:  Limited inclusion of qualitative results and human-rater experiments: It would further improve the paper if the author(s) can provide more qualitative results that demonstrates the differences between the proposed method and existing SoTA, and preferably include a lightweight human-rating study given that the metrics might not fully reflect human preference for layout generation.
>
> A: Thank you for recommending a lightweight human-rating study. We acknowledge its significance and aim to incorporate it in the final version of our paper. We have included demo qualitative results in Appendix C, comparing our model with LayoutDM and real data, which could be used for future human-rating studies. However, due to the time constraints of the rebuttal process, it is hard for us to complete the participant recruitment and execution of the study in time.
>
> W3: Missing common metrics: While FID and MaxIoU is quite representative and established in the layout generation literature, it will be helpful if the author(s) can include alignment and overlap indices, which can further show the effect of the proposed aesthetic constraints.
>
> A: Thank you for your suggestion. As mentioned in our global response, we have incorporated the requested alignment and overlap indices results in Appendix A. We believe these additional results have further enhanced the quality of our paper.
>
> Q2: Will the code be open-sourced if this paper is accepted?
>
> A: Yes, we plan to include a GitHub link of the source code and trained checkpoints in the final version of the paper. And we have provided source code through an Anonymous GitHub link in the revised version.
>
> Q3: Can you further discuss the importance of the time-dependent constraint weight (preferably empirically)? This appears to be a very important parameter in incorporating aesthetic constraints.
>
> A: The time-dependent constraint weight is critical for effective model convergence and output quality. Without this weight, the model struggles to converge, leading to a high Fréchet Inception Distance (FID) score, typically remaining above 100, which indicates poor layout quality. We determine the weight empirically as explained in Appendix B.1.

---

> > ### Comment · Reviewer_6Y7v · 2023-11-22
> >
> > Thank you for addressing my concerns. My rating remains unchanged and I am happy to support the acceptance of this paper.

---

### Official Review · Reviewer_DJP7 · 2023-11-03

**Soundness:** 3 good
**Presentation:** 3 good
**Contribution:** 2 fair
**Rating:** 6
**Confidence:** 3

**Summary:**

The paper proposes a model for unconditional and conditional layout generation, and layout de-noising, where layout is defined as a set of rectangles of particular clas placed on the larger canvas, and conditioning can take form of number of rectangles of each class.The main contribution of the paper is modelling this process as continuous diffusion process, as opposed to the previous work which did it as a discrete diffusion process, after bining the rectangular coordinates. The method outperforms other layout generation models, as well as generic models such as MaskGIT, as measured by FID score and the IoU-related metric. The additional contribution is the introduction of a new aesthetic constraint, which further improves the results, used both as a loss during training, and during post-processing, where the rectangles are adjusted to reduce the object overlap.

**Strengths:**

Originality: While neither the method nor the application domain is new, it is novel to apply the continuous diffusion to the layout generation problem.
Clarity: The writing is sufficiently clear.
Quality: The paper features an ablation study highlighting the most important design decisions, such as post-processing, and the use of aesthetic constraint to minimize the loss. The authors promise to make the code and model checkpoints available, but do not provide them together with the submission.
Significance: I believe the work has significance in exploring the use of continuous diffusion to layout generation task, but given the somewhat "toy" setup of the experiments, the practical applicability and the wider significance of layout generation when specified in this formulation (given only types of objects or initial layout, without any information about the textual content of text fields, or image contents of the image fields, with evaluation done through FID and not human studies) may be limited.

**Weaknesses:**

There aren't significant drawbacks in the manuscript itself, with main potential weakness for me being the question of practicality of the proposed approach or its extensions to any practical application.

**Questions:**

Do you envision any practical applications of the proposed approach? Do you believe it could be extended in a way that would be useful for applications? In which way?

Sec. 2.2, "Conditional generation": Typo "We" --> "we"
Page 4, section "Overlap constraint": Typo "is monotonically identify" --> "monotonically identifies"
Page 6, Sec. 4.1, "Datasets": Typo "for both dataset" --> "for each dataset" or "for both datasets"

---

> ### Author Response · Authors · 2023-11-20
>
> W1: There aren't significant drawbacks in the manuscript itself, with main potential weakness for me being the question of practicality of the proposed approach or its extensions to any practical application.
>
> A: Thank you for your feedback. Regarding potential applications, please refer to our detailed response to Question 1.
>
> Q1: Do you envision any practical applications of the proposed approach? Do you believe it could be extended in a way that would be useful for applications? In which way?
>
> A: Layout generation can facilitate the design of posters, websites, presentation slides, and mobile applications. For example, conditional layout generation can be applied to convert text files and images into well-organized reports. Additionally, unconditional generation can be integrated with large language models (like ChatGPT) and image generation models (such as Stable Diffusion) for complete automation, with text and image generation models filling empty layout elements.
>
> In addition, layout generation could be used to assist other generative tasks, including layout-guided image generation [1] and visual text rendering [2]. However, future work is required to enable visual and text content awareness in layout generation.
>
> [1] Li, Yuheng, et al. "Gligen: Open-set grounded text-to-image generation." Proceedings of the IEEE/CVF Conference on Computer Vision and Pattern Recognition. 2023.
>
> [2] Chen, Jingye, et al. "TextDiffuser: Diffusion Models as Text Painters." arXiv preprint arXiv:2305.10855 (2023).
>
> Q2: Sec. 2.2, "Conditional generation": Typo "We" --> "we" Page 4, section "Overlap constraint": Typo "is monotonically identify" --> "monotonically identifies" Page 6, Sec. 4.1, "Datasets": Typo "for both dataset" --> "for each dataset" or "for both datasets"
>
> A: Thank you for pointing out these errors. We have carefully corrected them in our manuscript. Additionally, we have conducted further proofreading to ensure its overall accuracy and clarity.

---

### Author Response · Authors · 2023-11-20
**Global Response**

We thank the reviewers for their valuable comments. We are happy that the reviewers find our work interesting and well-written in general. But there are also some misunderstandings and concerns from several perspectives. A common issue raised by the reviewers is the lack of the overlap metric in our evaluation, which we will address below. For other specific questions raised by each reviewer, we will post our responses separately. We also have added requested results in the reviews in the Appendix and added reference in the main text. We will incorporate more detailed revisions into the camera-ready version according to our responses to the reviews.

Q: Missing common metrics: While FID and MaxIoU is quite representative and established in the layout generation literature, it will be helpful if the author(s) can include alignment and overlap indices, which can further show the effect of the proposed aesthetic constraints.

A: Thank you for the suggestion. We have included overlap metric results and discussion in Appendix A. We compare our model with LayoutDM, a discrete diffusion model with the same network backbone, and LayoutGAN++, which also employs beautification constraints and post-processing. We used the overlap metric from LayoutGAN++ and its implementation from LayoutDM's source code. The results demonstrate the efficacy of the overlap constraint and also proved our method is more effective in incorporating the constraints than LayoutGAN++.

We did not show the overlap metric in the initial submission, mainly for two reasons:

1. The overlap metric is unsuitable for assessing model quality trained on the Rico dataset, where layout overlap is not supposed to be avoided.
2. Existing methods present unclear and inconsistent definitions of the overlap metric without available implementations. e.g., LDGM claims to use the unnormalized overlap metric [1]. Yet, the magnitude of their reported overlap scores does not align with those in the original publication, and the scale parameter is unspecified. Similarly, He, Liu, et al. [2] employ a percentage-based overlap metric. However, they did not explain their approach to addressing asymmetry in the denominator, nor have they shared their source code.

[1]. Li, Jianan, et al. "Attribute-conditioned layout gan for automatic graphic design." IEEE Transactions on Visualization and Computer Graphics 27.10 (2020): 4039-4048.

[2]. He, Liu, et al. "Diffusion-based document layout generation." arXiv preprint arXiv:2303.10787 (2023).

---

### Author Response · Authors · 2023-11-22
**Rebuttal Deadline Follow-Up**

As the rebuttal deadline approaches, I would like to thank you for your constructive feedback. We have diligently addressed your concerns in our revisions. We believe these improvements have strengthened the paper, and we hope these enhancements align more closely with the conference's standards. We appreciate your consideration of our updated submission and look forward to any further suggestions you may have.

---

### Meta-Review · Area_Chair_C4AW · 2023-12-08

**Metareview:**

The paper proposes an approach for layout generation based on continuous diffusion models. The work supports a variety of layout generation tasks and instruments "aesthetic constraints" in training. This is interesting work. There are a few major issues that the authors should address in the revision. First of all, the paper has missed a closely related work, "Play: Parametrically Conditioned Layout Generation using Latent Diffusion" by Chin-Yi Cheng et al. ICML 2023. The work uses latent diffusion instead of directly operating in the layout space. The authors should discuss the work in the revision. In addition, the author should discuss how the FiD scores reported in this submission different from the one used in Play that measures visual similarity.

The reviewers think the work applies continuous diffusion in an interesting scenario. Reviewer DJP7 mentioned that "While neither the method nor the application domain is new, it is novel to apply the continuous diffusion to the layout generation problem." Reviewer 6Y7v found the formulation of Overlap and Alignment losses and time-dependent constraint scheduling novel. On the negative side, the most important limitation seems to be the lack of human evaluation. All the results are reported based on automatic metrics. The reviewers brought up a number of other issues regarding the experiments / ablation and clarity. The reviewers seem happy with the author responses.

**Justification For Why Not Higher Score:**

The paper suffer a few issues as mentioned above.

**Justification For Why Not Lower Score:**

The work is making a nice contribution for layout generation.

---

### Decision · Program_Chairs · 2024-01-16

Accept (poster)